# Optimization of the Fermentation Conditions of *Metarhizium robertsii* and Its Biological Control of Wolfberry Root Rot Disease

**DOI:** 10.3390/microorganisms11102380

**Published:** 2023-09-23

**Authors:** Jing He, Xiaoyan Zhang, Qinghua Wang, Nan Li, Dedong Ding, Bin Wang

**Affiliations:** College of Forestry, Gansu Agricultural University, Lanzhou 730070, China; zxy1039507106@163.com (X.Z.); wqh1231212@163.com (Q.W.); lilinannan0709@163.com (N.L.); ddd3202022@163.com (D.D.); wangbin@gsau.edu.cn (B.W.)

**Keywords:** endophytic fungus, *Lycium barbarum*, disease control, *Fusarium solani*, antifungal mechanism

## Abstract

*Fusarium solani* is the main pathogenic fungus causing the root rot of wolfberry (*Lycium barbarum*). The endophytic fungus *Metarhizium robertsii* has been widely used for the biocontrol of plant pathogenic fungi, but the biocontrol effects of this fungus on wolfberry root rot and its antifungal mechanism against *F. solani* have not been reported. In this study, the antagonism of endophytic fungus *M. robertsii* against *F. solani* was verified. Further, we optimized the fermentation conditions of *M. robertsii* fermentation broth based on the inhibition rate of *F. solani*. In addition, the effects of *M. robertsii* fermentation broth on the root rot of wolfberry and its partial inhibition mechanism were investigated. The results showed that *M. robertsii* exhibited good antagonism against *F. solani*. Glucose and beef extracts were the optimal carbon and nitrogen sources for the fermentation of *M. robertsii*. Under the conditions of 29 °C, 190 rpm, and pH 7.0, the fermentation broth of *M. robertsii* had the best inhibition effect on *F. solani*. Furthermore, the fermentation broth treatment decreased the activities of superoxide dismutase, catalase, and peroxidase of *F. solani*; promoted the accumulation of malondialdehyde; and accelerated the leakage of soluble protein and the decrease in soluble sugar. In addition, inoculation with *M. robertsii* significantly reduced the decay incidence and disease index of wolfberry root rot caused by *F. solani*. These results indicate that *M. robertsii* could be used as a biological control agent in wolfberry root rot disease management.

## 1. Introduction

As a medicinal and functional food, wolfberry (*Lycium barbarum*) has a long history of planting and cultivation, and it is widely planted in the Nei Monggol, Gansu, Ningxia, Shaanxi, and Qinghai provinces in China [1]. This plant has a high nutritional value and contains a variety of bioactive compounds such as polysaccharides, minerals, carotenoids, and polyphenols, for which their various effects include anticancer, antiaging, and hypoglycemic [2]. However, wolfberry plants are susceptible to phytopathogenic fungi, resulting in decreased fruit quality and yield. Among them, root rot caused by *Fusarium solani* is one of the major soil-borne diseases. The disease occurs in a wide range, spreads rapidly, and is highly destructive. It can cause the yellowing of plant leaves and the shrinking of branches, resulting in a decline in the quality of wolfberry fruit and a decrease in yield. In severe cases, it can lead to the death of the entire plant, causing great economic losses to local wolfberry growers [3]. Therefore, the effective prevention and control of root rot disease are of great significance for the healthy development of the wolfberry industry.

Biological control agents have been widely used to control plant root rot due to their advantages of safety, high efficiency, and low cost. *Trichoderma harzianum*, *Rhizobium japonicum*, and *T. atroviridae* treatments significantly reduce the incidence and severity of peanut and soybean root rot caused by *F. solani*, and they also show good plant growth promotion effects [4,5,6]. *Metarhizium robertsii* is a common entomopathogenic fungus and has been proven to be a plant endophytic fungus [7,8,9]; it has a significant and persistent pest control effect on its natural insect enemies. Metarhizium biological agents have been commercialized to some extent in the United States, Brazil, and Europe. The application of *M. robertsii* showed good biological control against banana stem weevil *Odoiporus longicollis* Oliver and European grapevine moth *Lobesia botrana* [10,11]. In addition, *Metarhizium robertsii* also showed good effects in plant disease control and significantly decreased the disease index of soybean root rot caused by *F. solani* [7]. Antioxidant systems play an important role in ROS scavenging, and superoxide dismutase (SOD), catalase (CAT), and peroxidase (POD) are important antioxidant enzymes in pathogenic fungi [12]. The inhibition of antioxidant enzyme activity may disrupt the balance of ROS metabolism, thus affecting the growth and pathogenicity of pathogenic fungi. Malondialdehyde (MDA) is one of the indicators used to measure the degree of oxidative stress, which can reflect the degree of fungal membrane lipid peroxidation. Soluble protein and soluble sugar could act as measures for the level of protein damage and cell carbon metabolism. Our previous study found that *M. robertsii* is also an endophytic fungus of wolfberry, but its control effect on the root rot of wolfberry has not been reported. In addition, whether its antifungal mechanism is related to the reduction in antioxidant enzymes and the destruction of cell membrane structure remains unclear.

The production of secondary metabolites with antifungal effects is the key to biological control. The production of these antifungal substances is not only related to the genetic characteristics of the fungus itself but is also influenced by the medium, nutrient composition, and fermentation conditions. The optimization of medium composition and fermentation conditions can significantly increase the production of antifungal secondary metabolites and enhance the inhibitory effect on pathogenic fungi [13]. In one study, corn steep liquor as a nitrogen source promoted the growth of *M. robertsii* blastospore and increased its virulence relative to corn leafhopper *Dalbulus maidis* [14]. The optimal fermentation conditions obtained using response surface methodology optimization significantly increased the inhibition of *Fulvia fulva* and *Botryosphaeria dothidea* by *Streptomyces lavendulae* fermentation broth [15]. Although *M. robertsii* has been reported to inhibit the root rot of soybeans, the relationship between this fungus’s fermentation conditions and its biocontrol potential against *F. solani* is still unclear. 

The aims of this study were to (1) analyze the effects of different carbon and nitrogen sources as well as fermentation conditions on the antifungal activity of *M. robertsii*; (2) optimize the fermentation conditions of *M. robertsii* using response surface methodology to increase the antifungal activity of the fermentation broth; and (3) investigate the biocontrol effect of *M. robertsii* on the root rot of wolfberry and possible antifungal mechanism against *F. solani*.

## 2. Materials and Methods

### 2.1. Fungal Strain, Culture Medium, and Wolfberry Plant

The isolation, screening, identification, and preservation of *M. robertsii* (HYC-7 strain) and *F. solani* were carried out at the Forest Protection Laboratory, College of Forestry, Gansu Agricultural University. 

Basic fermentation medium: NaNO_3_ (3 g), KH_2_PO_4_ (1 g), MgSO_4_ (0.5 g), KCl (0.5 g), FeSO_4_ (0.01 g), sucrose (30 g), potato (200 g), and distilled water (up to 1 L). 

One-year-old healthy wolfberry plants were collected from the economic forest teaching and research practice base of Gansu Agricultural University.

### 2.2. Determination of the Antagonistic Effect of HYC-7 Strain on F. solani

*Fusarium solani* was inoculated on one side of the PDA plate, and the HYC-7 strain was inoculated on the other side at a symmetrical position 3 cm from the edge of the plate. After culturing at 25 °C for 5 d, the antagonistic effect was observed, and the inhibition rate of *F. solani* was calculated: inhibition rate = (colony diameter of control group − colony diameter of treatment group)/(colony diameter of control group) × 100%.

### 2.3. Screening of Optimum Carbon and Nitrogen Sources for Fermentation Medium

Mannitol, glucose, maltose, lactose, and soluble starch were selected as carbon sources to replace sucrose; beef extract, yeast extract paste, L-glutamic acid, carbamide and peptone (purity ≥ 99.7%, Tianjin Guangfu Fine Chemical Research Institute, Tianjin, China) were selected as nitrogen sources to replace sodium nitrate in the basic medium. 

Different nitrogen and carbon source liquid fermentation media were each placed into 150 mL conical flasks for 30 min. After cooling, 1 mL of the spore suspension (1 × 10^7^ mL^−1^) of the HYC-7 strain was added to each fermentation liquid medium; sterile water and Tween-80 were selected as the control. The medium was incubated in a constant-temperature shaker at 28 °C and 160 rpm for 5 d. The fermentation broth was centrifuged at 4 °C and 9900× *g* for 20 min to obtain the supernatant and then filtered through a 0.22 μm microporous membrane for later use. The fermentation broth was mixed with the PDA medium at a ratio of 30% [8]. After cooling, the fungus cake of *F. solani* was inoculated in the center of the plate and cultured at 28 °C in the dark. When the colony diameter in the control group reached 3/4 of the diameter of the plate, it was measured using the crossover method and the inhibition rate was calculated according to the following formula:

Inhibition rate (%) = $\frac{(A-B)}{\left(A\right)}\;$× 100; A and B represent the colony diameter of the control and treatment group, respectively. 

### 2.4. Single-Factor Test of the Effect of Different Fermentation Conditions on the Inhibition Rate of HYC-7 Fermentation Broth

Under the basic conditions of temperature (28 °C), pH (6.0), inoculation amount (1 mL), loaded liquid (60 mL), and rotational speed (160 rpm), we kept the other conditions constant and only changed one condition to conduct a single-factor test. The following conditions were set: temperature: 20, 22, 25, 28, and 30 °C; pH: 5.0, 6.0, 7.0, 8.0, 9.0, and 10.0; inoculation amount: 0.5, 1, 1.5, 2, 2.5, and 3 mL; loaded liquid: 40, 50, 60, 70, and 80 mL; rotational speed: 120, 140, 160, 180, and 200 rpm. Six replicates of each treatment were created and incubated for 5 d in a constant-temperature shaker; then, the inhibition rate was determined using the fermentation broth according to the method in Section 2.3.

### 2.5. Response Surface Optimization Test

On the basis of the single-factor test, the three pH (A), rotational speed (B), and temperature (°C) parameters were optimized. The response surface test scheme was designed according to the Box–Behnken method in the Design-Expert 8.0.6 software, as shown in Appendix A. 

### 2.6. Determination of Colony Diameter, Sporulation, Spore Germination Rate, and Germ Tube Length

The HYC-7 strains were inoculated in a basic 60 mL fermentation medium and cultured in a constant-temperature shaker at 190 rpm and 29 °C for 5 d. Then, the fermentation broth was centrifuged at 4 °C and 9900× *g* for 20 min, and the supernatant was taken and filtered through a 0.22 μm microporous membrane. A sterile fermentation filtrate was obtained for use.

The colony diameter, sporulation, spore germination rate, and germ tube length were determined according to the method of Li et al. (2020) [16]. The sterile fermentation filtrate was mixed with the PDA medium according to volume fractions of 10%, 20%, 30%, 40%, and 50%, and the basic fermentation medium was used as the control. The fungus cake of *F. solani* was inoculated to the center of the plate at 28 °C in the dark for 7 d. The colony diameter was measured using the crossover method. A plate cultured for 7 d was taken, and 10 mL of sterile water was added. Spores were gently scraped off the plate with a sterilizing coater. After the sterile gauze was filtered, sporulation was counted using a hemocytometer.

The sterile fermentation filtrate was mixed with the PDB medium in volume fractions of 10%, 20%, 30%, 40%, and 50%, and the same amount of basic fermentation medium was added as the control. With a pipette gun, the medium was absorbed and suspended on a hollow glass slide and cultured at 28 °C in the dark. After 6 h, a microscope was used to count the spore germination and measure the length of the germ tubes. Germ tubes longer than half of the spore length were considered as spore germination.

### 2.7. Determination of Antioxidant Enzymes Activity

After *F. solani* were cultured in PDB medium for 7 d, they were filtered and collected; washed with sterile water; mixed with fermentation broth at 1:50 (*w*/*v*); treated for 0, 3, 6, 9, 12, 24, 36, and 48 h; and stored in liquid nitrogen for later use. 

The assays of SOD, CAT, and POD activities were based on the methods previously described by Zhang et al. (2022) and Wang et al. (2021) [17,18]. In total, 0.2 g of frozen mycelium was ground in 5 mL of precooled phosphate buffer (50 mM and pH 7.8) and centrifuged at 4 °C and 9900× *g* for 20 min, and the supernatant was the crude enzyme of SOD and CAT. The SOD determination reaction system included 1.5 mL of phosphate buffer (50 mM, pH 7.8), 0.3 of mL L-methionine (130 mM), 0.3 mL nitroblue tetrazolium chloride (750 μM), 0.3 mL of EDTA-Na_2_ (100 μM), 0.5 mL of distilled water, 0.3 mL of riboflavin (20 μM), and 100 μL of crude enzyme solution. Subsequently, the mixture was placed in 25 °C and 3000–4000 lx light conditions for 20 min and then placed in the dark to terminate the reaction; the absorbance value was immediately measured at 560 nm. One activity unit (U) of the SOD enzyme inhibits 50% of the photochemical reduction of NBT and is expressed as U·g^−1^ FW. The CAT determination reaction system included 2.9 mL of H_2_O_2_ (0.067%) and 0.1 mL of enzyme solution, with distilled water as the control. The change in OD_240_ within 2 min was recorded. A decrease of 0.01 OD_240_ per minute was defined as one unit (U) and expressed as U·g^−1^ FW. The POD determination reaction system included 2.6 mL of guaiacol (0.3%), 0.3 mL of H_2_O_2_ (0.6%), and 0.1 mL of enzyme solution, with distilled water as the control. The change in OD_470_ within 2 min was recorded. An increase of 0.01 OD_470_ per minute was defined as one unit (U) and expressed as U·g^−1^ FW.

### 2.8. Determination of Content of Malondialdehyde, Soluble Protein, and Soluble Sugar

Malondialdehyde content was determined according to the method of Li et al. (2020) [15]. The 0.2 g of frozen mycelium was ground in 5 mL of TCA (10%) in an ice bath and centrifuged at 4 °C and 9900× *g* for 20 min, and the supernatant was the crude extract. In total, 1 mL of crude extract was added to 2 mL of 0.67% thiobarbituric acid in a boiling water bath for 30 min and then centrifuged after rapid cooling. The absorbance value of the supernatant was measured at 450 nm, 532 nm, and 600 nm. The MDA content was expressed as mmol·g^−1^ FW.

The soluble protein content was determined according to the method of Bradford. (1976) [19]. Then, 0.2 g of frozen mycelium was ground in 5 mL of distilled water in an ice bath and centrifuged at 4 °C and 9900× *g* for 20 min; the supernatant was a protein extraction solution. An amount of 0.5 mL of the extract was added to 0.5 mL of distilled water and 5 mL of Coomas bright blue G-250 reagent. After standing for 2 min, the absorbance value was measured at 595 nm. The soluble protein content was calculated using a standard curve and expressed as mg·g^−1^ FW. 

The content of soluble sugar was determined according to the method of Dai et al. (2017) [20]. Then, 0.2 g of frozen mycelium was ground in 5 mL of distilled water and transferred to a test tube, boiled for 30 min, naturally cooled, and filtered into a 25 mL volumetric flask. In total, 0.5 mL of filtrate was added to 1.5 mL of distilled water, 0.5 mL of anthrone solution, and 5 mL of concentrated sulfuric acid successively. After the reaction solution became transparent, the absorbance value was determined at 620 nm. The soluble sugar content was expressed as mg·g^−1^ FW.

### 2.9. Determination of Decay Incidence and Disease Index of Wolfberry

Healthy wolfberry root tissues were selected and washed with running water to remove soil. Then, they were soaked in 75% alcohol for 20 s, 0.1% mercuric chloride solution for 30 s, and finally rinsed with sterile water to remove the disinfectant residue. The roots were cut into 10 mm slices with a sterile blade, and the surface of the roots was pierced evenly with a sterile needle. HYC-7 strain, *F. solani,* and HYC-7 strain + *F. solani* were inoculated as treatment groups, and sterile water was used as the control. The specific experimental steps were as follows: The injured wolfberry root tissues were soaked in the fermentation broth of the HYC-7 strain with a concentration of 1 × 10^7^ spores/mL for 30 min and then placed in a Petri dish with moist sterile filter paper. *Fusarium solani* with a concentration of 1 × 10^6^ spores/mL were sprayed uniformly on the surface of the root tissues after 24 h and then cultured at 28 °C in the dark. According to the severity of the root’s decay, they were divided into five grades, where 0 = no root rot symptoms; 1 = less than 5% root area rotted; 2 = 5–25% root area rotted; 3 = 26–50% root area rotted; 4 = 51–75% root area rotted; and 5 = more than 75% root area rotted. After 7 d, the decay incidence and disease index were determined according to the method of Sasan and Bidochka. (2013) and calculated as the following formulas [7].
$$\mathrm{Decay}\;\mathrm{incidence}\;(\%)=\frac{\mathrm{Number}\;\mathrm{of}\;\mathrm{diseased}\;\mathrm{plants}}{\mathrm{Investigation}\;\mathrm{of}\;\mathrm{total}\;\mathrm{number}\;\mathrm{of}\;\mathrm{plants}}\times 100$$
$$\mathrm{Disease}\;\mathrm{index}\;(\%)=\frac{\Sigma (\mathrm{Number}\;\mathrm{of}\;\mathrm{disease}\;\mathrm{at}\;\mathrm{all}\;\mathrm{levels}\times \mathrm{Disease}\;\mathrm{grade})}{\mathrm{Survey}\;\mathrm{the}\;\mathrm{total}\;\mathrm{number}\;\mathrm{of}\;\mathrm{plants}\times \mathrm{Highest}\;\mathrm{grade}}\times 100$$

### 2.10. Data Analysis

All determinations in this study were repeated at least three times. Data were expressed as means and standard errors, and Origin 2022b was used for mapping. The significance analysis of Duncan’s multiple differences was performed using SPSS 22.0 (SPSS, Chicago, IL, USA).

## 3. Results

### 3.1. HYC-7 Strain on F. solani

The results of the PDA plate confrontation showed that the HYC-7 strain had a significant antagonistic effect on *F. solani*, with an inhibition rate of 39. 8% at the 5th d of culture (Figure 1).

### 3.2. Screening of Optimum Nitrogen and Carbon Sources for Fermentation Medium

Nitrogen and carbon sources are important nutrients for fungal growth. The results showed that the inhibition rate of the HYC-7 fermentation broth was significantly different in different nitrogen and carbon source media. When beef extract and glucose were used as nitrogen and carbon sources, the inhibition rates of the fermentation broth were 40.35% and 35.58%, respectively, which were significantly higher than those of other nitrogen and carbon sources (*p* < 0.05) (Figure 2). Therefore, the fermentation medium with beef extract as the nitrogen source and glucose as the carbon source was selected for the single-factor test and the collection of HYC-7 strain fermentation broth.

### 3.3. Single-Factor Test 

The inhibition rate of the HYC-7 fermentation broth showed a first increasing and then decreasing trend with an increase in pH. When the pH was 9, the highest inhibition rate was 38.9% (Figure 3A). When the inoculation amount was within the range of 0.5–3.0 mL, the inhibition rate of the HYC-7 fermentation broth showed an obvious first increasing and then decreasing trend. When the inoculation amount was 2 mL, the inhibition rate was 28.11%, which was significantly higher than other groups (*p* < 0.05) (Figure 3B). The loaded liquid had a significant effect on the inhibition rate of the HYC-7 fermentation broth. When the loaded liquid was 70 mL, the maximum inhibition rate was 30.75% (Figure 3C). The inhibition rate of the fermentation broth increased with an increase in rotational speed. The maximum inhibition rate was 38.39% at 200 rpm (Figure 3D). The inhibition rate of the fermentation broth decreased first and then increased with an increase in temperature and reached a maximum value of 38.02% at 28 °C (Figure 3E).

### 3.4. Response Surface Test Optimization Results

The response surface test’s design and results are shown in Appendix A.

#### 3.4.1. Regression Equation Fitting and Analysis of Variance

A mathematical model with the regression equation was established via statistical analyses of the experimental data: the inhibition rate (%) = 51.71 − 3.09 A − 0.10 B + 3.37 C + 2.96 AB + 0.45AC + 0.80 BC − 4.38 A^2^ − 8.39 B^2^ − 7.51 C^2^.

Variance analysis and the significant difference test were conducted for the regression model, and the results are shown in Appendix A. The regression of the model was significant (*p* < 0.0001). The loss of quasi-item *p* = 0.1179 > 0.05 was not significant, indicating that the model was suitable with a high degree of fit, indicating that the test results were consistent with the model, while other unknown factors had little interference in the test results; the model was suitable, and the fitting degree was high, so the establishment of the regression model had a certain guiding significance. At the same time, the first A, C, and AB terms and the second A^2^, B^2^, and C^2^ terms all had significant antifungal activity. The correlation of the model was high with a regression coefficient of R^2^ = 0.9777. The regression coefficient was R^2^ = 0.9777; this showed that the correlation of the model was high. The F value represents the importance of each influencing factor to the test index. The larger the F value, the stronger the influence of the factor on the inhibition rate. The results showed that the influence of three factors on the inhibition rate was in the order of temperature (C, °C) > pH (A) > rotational speed (B, rpm).

#### 3.4.2. Response Surface Analysis of Interaction of Various Factors

The response surface diagram below could more intuitively reflect the interaction of the three main factors and their influence on the inhibition rate. It can be observed in the surface diagram and contour lines in (Figure 4) that the interaction between A (pH) and B (rotational speed) had a significant impact on the antifungal activity of the strain. This is consistent with the results shown in Appendix A, such as AB = 0.0102 < 0.05. The optimal conditions were obtained using a quadratic multinomial regression fitting equation: pH—6.96; rotational speed—189.40 rpm; and fermentation temperature—29.21 °C. Under these conditions, the predicted inhibition rate was 52.62%. The optimal fermentation conditions were a pH of 7.0; a rotational speed of 190 rpm; and a culture temperature of 29 °C for simple and feasible operation.

The fermentation broth was prepared under the optimal fermentation conditions (pH 7.0, 190 rpm, and 29 °C) and repeated three times to verify the accuracy of the model. The average inhibition rate was 51.80%, which was consistent with the predicted value of 52.62%.

### 3.5. Effects of HYC-7 Fermentation Broth on the Growth and Development of F. solani

The colony diameter could visually reflect the amount of mycelium growth. Compared with the control, the colony diameter of *F. solani* decreased continuously with the increase in fermentation broth concentration. When the concentration of fermentation broth was 50%, the colony diameter was the smallest, which was 65.2% lower than the control (*p* < 0.05) (Figure 5A). Sporulation, spore germination rate, and germ tube length are important indicators for measuring fungal reproductive capacity and spore viability. Compared with the control, the lowest sporulation was observed when the fermentation broth concentration was 30%, which was 82.1% lower than the control (*p* < 0.05) (Figure 5B). When the fermentation broth concentration was 30%, 40%, and 50%, there were no significant differences in sporulation. Similarly, both the spore germination rate and germ tube length of *F. solani* gradually decreased with the increase in fermentation broth concentration. When the concentration of the broth was 50%, the spore germination rate and germ tube length were the lowest, and they were 75.8% and 62.6% lower than the control (*p* < 0.05) (Figure 5C,D). 

### 3.6. Effects of HYC-7 Strain Fermentation Broth on the Activities of SOD, CAT, and POD and the Contents of MDA, Soluble Protein, and Soluble Sugar 

SOD, CAT, and POD are important antioxidant enzymes in phytopathogenic fungi and play an important role in the scavenging of excess reactive oxygen species (ROS). The SOD activity of both the HYC-7 fermentation treatment and the control *F. solani* increased and then decreased during the treatment period, reaching a peak at 24 h. The HYC-7 fermentation broth treatment significantly decreased the SOD activity of *F. solani* and was lower than the control during the treatment period. At 24 h and 48 h, they were 29.89% and 64.01% lower than the control, respectively (*p* < 0.05) (Figure 6A). The activity of CAT in both the HYC-7 fermentation broth treatment and the control *F. solani* also increased and then decreased during the treatment period, reaching a peak at 24 h. The HYC-7 fermentation broth treatment significantly decreased the CAT activity of *F. solani* in the later treatment period, and it was lower than the control by 20.25% and 41.66% at 24 h and 48 h, respectively (*p* < 0.05) (Figure 6B). The POD activity of the control *F. solani* increased continuously during the treatment, but the HYC-7 fermentation broth treatment increased first and then decreased, always being lower than that of the control. At 24 h and 48 h, these two were 25.73% and 78.17% lower than the control (*p* < 0.05) (Figure 6C). 

The MDA content of both the HYC-7 fermentation broth treatment and control *F. solani* increased continuously during the treatment period. The HYC-7 fermentation broth treatment significantly increased the MDA content of *F. solani* and was higher than the control during the treatment period. At 24 h and 48 h, it was 13.15% and 32.15% higher than the control, respectively (*p* < 0.05) (Figure 6D). The soluble protein content of the HYC-7 fermentation broth treatment and control *F. solani* increased first and then decreased during the treatment period. The control reached the peak at 36 h, but the broth treatment reached the peak at 24 h and then began to decrease rapidly. At 48 h, it was significantly lower than the control: 30.07% (*p* < 0.05) (Figure 6E). Similarly, the soluble sugar content of the broth treatment and control *F. solani* decreased continuously during the treatment period. The HYC-7 fermentation broth accelerated the decrease in soluble sugar content, which was 28.23% and 40.95% lower than that of the control at 24 h and 48 h, respectively (*p* < 0.05) (Figure 6F). 

### 3.7. Effect of HYC-7 Fermentation Broth on the Decay Incidence and Disease Index of Wolfberry Root Rot

As shown in Figure 7, the control group also showed a certain degree of decay incidence. Compared with the control, HYC-7 strain inoculation significantly decreased the decay incidence and disease index, which were 38.5% and 25.5% lower, respectively, than those of the control after 7 d of inoculation (*p* < 0.05). *Fusarum solani* inoculation resulted in a more serious decay incidence of wolfberry root rot. Similarly, compared with the *F. solani* inoculation, HYC-7 strain+*F. solani* inoculation significantly decreased the incidence and disease index by 57.5% and 41.8%, respectively, after 7 d of inoculation (*p* < 0.05). 

## 4. Discussion

In this study, we found that the antifungal activity of *M. robertsii* fermentation broth was strongest when glucose and beef paste were used as carbon and nitrogen sources. This result is similar to the previous results obtained by optimizing the composition of the *Bacillus pumilus* fermentation medium [21]. It was also shown that the best carbon and nitrogen sources for *M. anisopliae* to produce the secondary metabolite destruxin were maltose and peptone, respectively, which is inconsistent with the results obtained in the present study possibly due to differences among the strains [22]. The response surface results indicated that temperature, pH, and rotational speed had a greater effect on the antifungal activity of *M. robertsii* fermentation broth. The optimal fermentation conditions were a temperature of 29 °C, a pH of 7.0, a rotational speed of 190 rpm, and 60 mL of loaded liquid; the inhibition rate was 51.80% under these conditions. Previous studies have shown that the evaluation of biocontrol effectiveness is based on the content of antimicrobial substances or the inhibition rate of the target pathogenic fungi [23]. Using the antifungal ability of the fermentation broth as a response variable, the response surface methodology optimized the optimal medium volume, initial pH, and fermentation temperature of the DS-R5 strain, which significantly improved the inhibition of its fermentation broth against *F. solani* [24]. Similarly, the optimization of the fermentation conditions of *Xenorhabdus nematophila* using the response surface methodology significantly improved its antibiotic activity [13]. 

In this study, it was shown that the HYC-7 strain, isolated from healthy wolfberry inter-root soil, had a significant biological control effect against root rot disease caused by *F. solani*. Previous studies have shown that the key to the biocontrol effect of microorganisms is the production of secondary metabolites, and the main metabolites in the fermentation broth of *M. anisopliae* are the nonribosomal cyclic peptides of destruxins. Destruxins exhibit a variety of acute toxic effects against insects. In addition, serinocyclin, subglutinol, and swainsonine were also identified in the secondary metabolites of *Metarhizium* spp. [25]. Serinocyclin A showed entomophagous activity as the exposed mosquito larvae to this compound exhibited abnormal swimming as they were unable to control the position of their heads. Swainsonine, as a mycotoxin, has the effect of inhibiting lysosomal α-mannosidase [26,27]. In this study, the results indicated that the HYC-7 fermentation broth treatment with different concentrations had a significant inhibitory effect on the growth and development of *F. solani*, and the inhibition increased with the increasing concentration of fermentation broth volume. This may be due to the increase in the concentration of antifungal secondary metabolites in the broth with respect to the increase in the concentration volume.

Pathogenic fungi are subjected to severe oxidative stress when infesting plants or under unfavorable environmental conditions, which can limit their colonization or normal growth and development [28]. However, such fungi have the ability to scavenge reactive oxygen species (ROS) to neutralize excess ROS from normal physiological processes or environmental stresses. SOD catalyzes the conversion of O_2_^-^ to H_2_O_2_, followed by the further decomposition of H_2_O_2_ to H_2_O and O_2_ by CAT, thereby reducing ROS-induced oxidative stress. POD acts synergistically with CAT and SOD to form an antioxidant enzyme system that is also involved in the detoxification of H_2_O_2_ [29]. Previous studies found that citral and cinnamaldehyde treatments significantly reduced SOD, CAT, and POD activities and inhibited the growth of *F. sambucinum*. Our study indicated that HYC-7 fermentation broth treatment significantly inhibited the SOD, CAT, and POD activities of *F. solani*. The inhibition of antioxidant enzyme activity may lead to the disruption of the balance of ROS metabolism and the massive accumulation of intracellular reactive oxygen species, which exacerbate cellular membrane lipid peroxidation and disrupted membrane integrity, resulting in a significant reduction in pathogenicity [28]. Previous studies have shown that when the cell membrane is destroyed by antibacterial substances, it will cause changes in permeability, resulting in a large accumulation of MDA and the release of macromolecular proteins in cells [30]. In this study, we found that the HYC-7 fermentation broth treatment promoted the accumulation of MDA and accelerated the leakage of soluble protein and the decrease in soluble sugar content. Therefore, we inferred that the HYC-7 fermentation broth caused serious damage to the cell membrane and the leakage of soluble protein and inhibited the carbon metabolism of *F. solani*, limiting its normal life metabolism and thus effectively inhibiting the growth of *F. solani*. This also indirectly supported the results that the HYC-7 fermentation broth treatment significantly inhibited the activities of antioxidant enzymes. These results were consistent with previous reports that *Bacillus pumilus* HR10 fermentation broth treatment promoted MDA accumulation and inhibited soluble protein and soluble sugar content in *Sphaeropsis sapinea* [31]. In summary, we suggest that the phenomenon of HYC-7 fermentation broth reducing the occurrence of wolfberry root rot is related to inhibiting the growth and development of *F. solani*, promoting MDA accumulation and accelerating the leakage of soluble protein and the decrease in soluble sugar content; however, the specific mechanism needs to be further studied.

## 5. Conclusions

In this study, the fermentation broth of *M. robertsii* had the best inhibitory effect on *F. solani* when glucose and beef extract were selected as carbon and nitrogen sources. The fermentation factors affecting the inhibitory effect of *M. robertsii* fermentation broth were temperature > pH > rotational speed, and the optimal fermentation conditions were a temperature of 29 °C, a pH of 7.0, a rotational speed of 190 rpm, and 60 mL of loaded liquid; the inhibition rate was 51.80% under these conditions. *Metarhizium robertsii* fermentation broth treatment inhibited colony growth, sporulation, spore germination, and germ tube elongation of *F. solani*. *Metarhizium robertsii* fermentation broth treatment also decreased the SOD, CAT, and POD activities of *F. solani*; promoted the accumulation of MDA; accelerated the leakage of soluble protein; and reduced the soluble sugar content. In addition, *M. robertsii* inoculation significantly decreased the decay incidence and disease index of wolfberry root rot. In conclusion, we believe that *M. robertsii* has a good control effect on the root rot of wolfberry and could be used for the development of biological control agents.

## Figures and Tables

**Figure 1 microorganisms-11-02380-f001:**
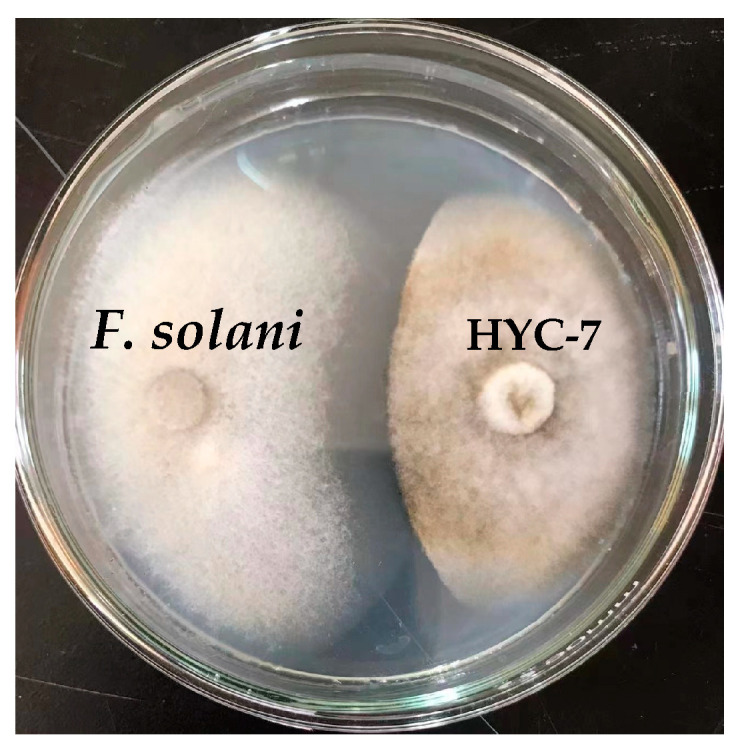
HYC-7 strain inhibits the colony growth of *F. solani*.

**Figure 2 microorganisms-11-02380-f002:**
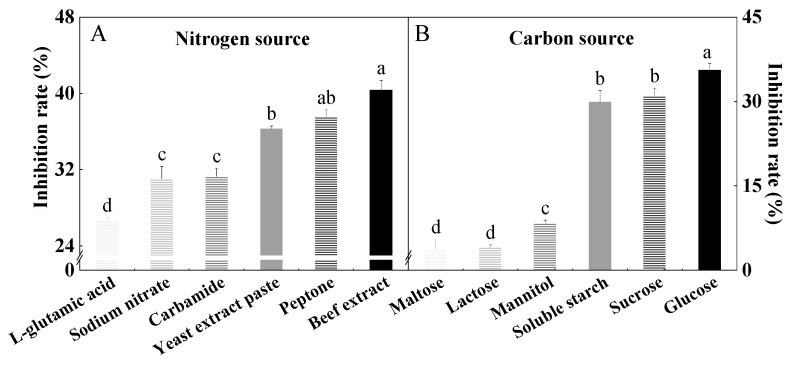
Effect of different nitrogen (**A**) and carbon sources (**B**) on the inhibition rate of HYC-7 strain fermentation broth. Vertical bars represent standard error (±SE), and different letters indicate significant differences between groups (*p* < 0.05).

**Figure 3 microorganisms-11-02380-f003:**
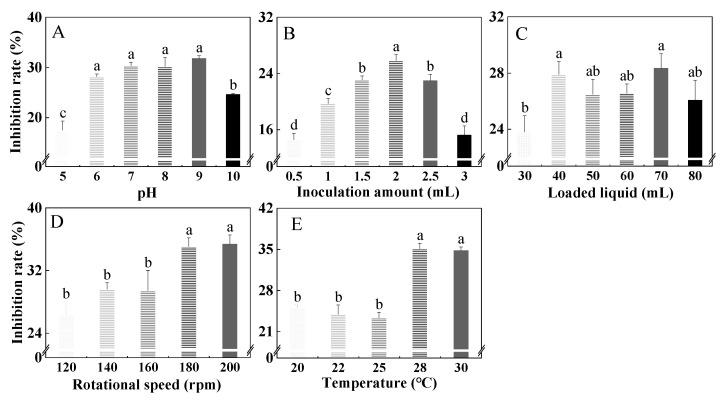
Effects of pH (**A**), inoculation amount (**B**), loaded liquid (**C**), rotational speed (**D**), and temperature (**E**) on the inhibition rate of the HYC-7 strain fermentation broth. Vertical bars represent standard error (±SE), and different letters indicate significant differences between groups (*p* < 0.05).

**Figure 4 microorganisms-11-02380-f004:**
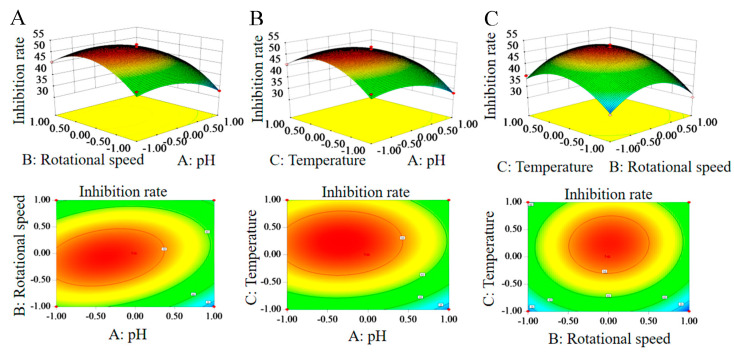
The response surface methodology and contour plots of the effects of the interaction between rotational speed and pH (**A**), temperature and pH (**B**), and temperature and rotational speed (**C**) on the inhibition rate of the HYC-7 strain fermentation broth. Note: The incline degree of the surface diagram is directly proportional to the influence degree of factors on the response value. The larger the focal length of the contour, the stronger the interaction between parameters.

**Figure 5 microorganisms-11-02380-f005:**
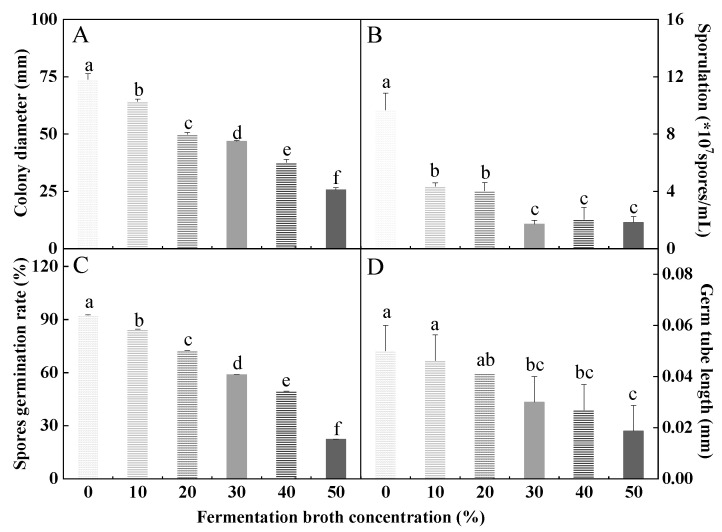
Effects of the HYC-7 strain fermentation broth with different concentrations on the colony diameter (**A**), sporulation (**B**), spore germination rate (**C**), and germ tubes length (**D**) of *F. solani*. Vertical bars represent standard error (±SE), and different letters indicate significant differences between groups (*p* < 0.05).

**Figure 6 microorganisms-11-02380-f006:**
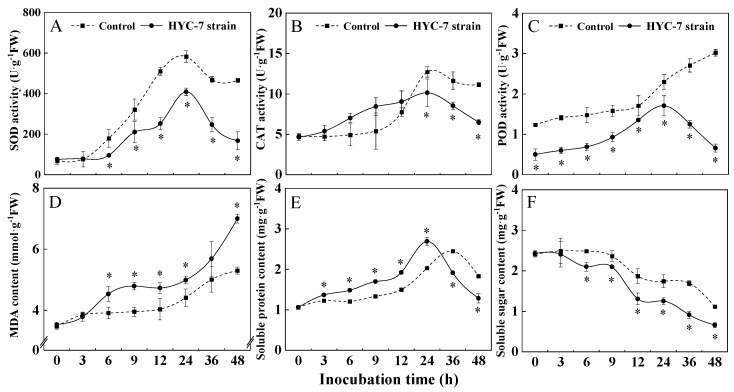
Effects of the HYC-7 strain fermentation broth on the activities of SOD (**A**), CAT (**B**), and POD (**C**) and the contents of MDA (**D**), soluble protein (**E**), and soluble sugar (**F**) of *F. solani*. Vertical bars represent standard error (±SE), and asterisks indicate significant differences between the treatment and the control at the same time (*p* < 0.05). SOD: Superoxide dismutase; CAT: catalase; POD: peroxidase; MDA: malondialdehyde.

**Figure 7 microorganisms-11-02380-f007:**
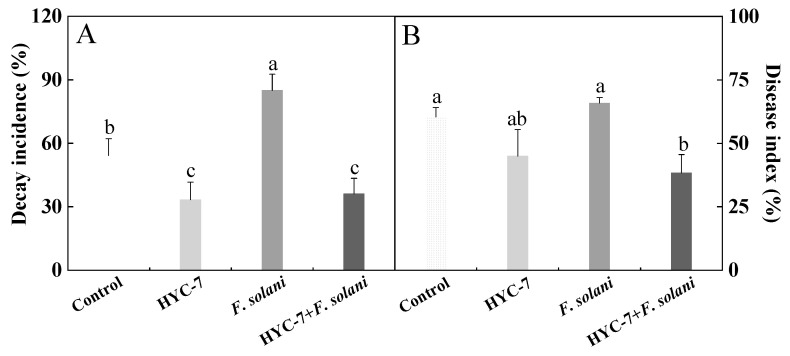
Effect of the HYC-7 strain fermentation broth on the decay incidence (**A**) and disease index (**B**) of wolfberry root rot. Vertical bars represent standard error (±SE), and different letters indicate significant differences between groups (*p* < 0.05).

## Data Availability

No new data were created or analyzed in this study. Data sharing is not applicable to this article.

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
