# Peer review of "Optimization of the Fermentation Conditions of Metarhizium robertsii and Its Biological Control of Wolfberry Root Rot Disease"

_microorganisms, 2023, doi:10.3390/microorganisms11102380_

Round 1

Reviewer 1 Report (Previous Reviewer 1)

Accept

Minor editing of English language required

Author Response

Thank you for your positive comments and valuable suggestions. We carefully examined the sentences to improve the accuracy and clarity of the manuscript. The manuscript has undergone English language editing by MDPI (english-67148), and the grammar, spelling and punctuation have all been updated and checked.

Reviewer 2 Report (Previous Reviewer 3)

The authors have made good progress on this manuscript and have corrected most of the past issues. Please review the below issues for better clarity.

Line 49: Consider changing to: on its natural insect enemies. 

Line 105: Consider replacing packed in with placed into

Line 235: Please consider stating directly what the next experiment is as it is a little vague.

Lines 257-259: This is extremely short for its own section as it is only one sentence with a heading and subheading. Consider combining this section with the 3.3 section. For example, 3.3 Single factor test and response surface test optimization results.

Lines 266-269: indicating is an interpretation and needs to be replaced with a result. Consider something like: ... the model was suitable with a high degree of fit.

Lines 271-273: The second part of this sentence starting with this showed is interpretation. Consider changing the order of the sentence: The correlation of the model was high with a regression coeffiecient R2=0.9777. Lines 272 and 273 should be removed or moved to the methods as they are not results but explaining the interpretation of the statistical method.

Line 389-391: The authors provide a list of secondary metabolites. It would be more informative to state what the function of some of these metabolites do.

There are a few sentences that needs some grammar review but it is greatly improved.

Author Response

Response to Reviewer 2 Comments

Comments and Suggestions for Authors

The authors have made good progress on this manuscript and have corrected most of the past issues. Please review the below issues for better clarity.

Response: Thank you for your positive comments and valuable suggestions.

Line 49: Consider changing to: on its natural insect enemies. 

Response 1: “on its natural enemy insects” has been changed to “on its natural insect enemies” in Line 79.

Line 105: Consider replacing packed in with placed into

Response 2: “packed in“ has been changed to “placed into” in Line 105.

Line 235: Please consider stating directly what the next experiment is as it is a little vague.

Response 3: We have changed the “next experiment“ to “single‐factor test and  collection of HYC-7 strain fermentation broth“.

Lines 257-259: This is extremely short for its own section as it is only one sentence with a heading and subheading. Consider combining this section with the 3.3 section. For example, 3.3 Single factor test and response surface test optimization results.

Response 4: We have made changes according to your suggestion.

Lines 266-269: indicating is an interpretation and needs to be replaced with a result. Consider something like: ... the model was suitable with a high degree of fit.

Response 5: We have revised the sentence as suggested below.

Line 266-267: The loss of quasi-item P=0.1179 > 0.05 was not significant, indicating that the model was suitable with a high degree of fit.

Lines 271-273: The second part of this sentence starting with this showed is interpretation. Consider changing the order of the sentence: The correlation of the model was high with a regression coeffiecient R2=0.9777. Lines 272 and 273 should be removed or moved to the methods as they are not results but explaining the interpretation of the statistical method.

Response 6: We have revised the sentence in Line 271-272 and removed the sentence in Line 272-273 as your suggestion.

Line 389-391: The authors provide a list of secondary metabolites. It would be more informative to state what the function of some of these metabolites do.

Response 7: We have revised the sentences as suggest below.

Destruxins exhibit a variety of acute toxic effects against insects. In addition, serinocyclin, subglutinol, and swainsonine were also identified in the secondary metabolites of Metarhizium spp.. Serinocyclin A showed entomophagous activity as the exposed mosquito larvae to this compound exhibited abnormal swimming as they were unable to control the position of their heads. Swainsonine, as a mycotoxin, has the effect of inhibiting lysosomal α-mannosidase[25].

Comments on the Quality of English Language

There are a few sentences that needs some grammar review but it is greatly improved.

Response 8: Thank you for your positive comments and valuable suggestions. We carefully examined the sentences to improve the accuracy and clarity of the manuscript. The manuscript has undergone English language editing by MDPI (english-67148), and the grammar, spelling and punctuation have all been updated and checked.

We have tried our best to improve the manuscript and have made some changes in the manuscript. We marked with red where we made the changes. These changes do not affect the content and scope of the manuscript. We hope that the correction we have made will meet this requirement.

Thank you for your consideration. I look forward to hearing from you soon.

With best regards,

Prof. Jing He

College of Forestry,

Gansu Agricultural University,

Lanzhou 730030,

China.

Reviewer 3 Report (New Reviewer)

Correct words

L135 - 190 rpm

L389, L370 – destruxin

I recommend adding after [25] references:

[26] Yadav, R.N.; Mahtab Rashid, M.; Zaidi, N.W.; Kumar, R.; Singh, H.B. Secondary metabolites of Metarhizium spp. and Verticillium spp. and their agricultural applications. In Secondary Metabolites of plant growth promoting Rhizomicroorganisms; Singh, H., Keswani, C., Reddy, M., Sansinenea, E., García-Estrada, C., Eds.; Springer: Singapore, 2019.

[27] Xu, Y-J.; Luo, F.; Li, B.; Shang, Y.;Chengshu Wang, Ch. Metabolic conservation and diversification of Metarhizium species correlate with fungal host-specificity. Frontiers in Microbiology, December 2016, doi:            10.3389/fmicb.2016.02020

L 394-395 Are there any known secondary metabolites of Metarhizium spp. with antifungal activity?

Author Response

Response to Reviewer 3 Comments

Comments and Suggestions for Authors

Correct words: L135 - 190 rpm; L389, L370 – destruxin

Response 1: We have corrected the words according to your suggestion.

I recommend adding after [25] references:

Yadav, R.N.; Mahtab Rashid, M.; Zaidi, N.W.; Kumar, R.; Singh, H.B. Secondary metabolites of Metarhizium spp. and Verticillium spp. and their agricultural applications. Secondary Metabolites of plant growth promoting Rhizomicroorganisms; Singh, H., Keswani, C., Reddy, M., Sansinenea, E., García-Estrada, C., Eds.; Springer: Singapore, 2019.

[27] Xu, Y-J.; Luo, F.; Li, B.; Shang, Y.;Chengshu Wang, Ch. Metabolic conservation and diversification of Metarhizium species correlate with fungal host-specificity. 

Frontiers in Microbiology, December 2016, doi: 10.3389/fmicb.2016.02020

Response 2: We have added the references according to your suggestion.

L 394-395 Are there any known secondary metabolites of Metarhizium spp. with antifungal activity ?

Response 3: At present, most of the studies on secondary metabolites of Metarhizium spp. are focused on insects. Although there are many references showing that Metarhizium spp. also has a good antagonistic effect on pathogenic fungi, most of them only prove that the fermentation broth or crude extract has antifungal activity, and there are few studies on the antifungal activity of clear secondary metabolites. Therefore, this is also the focus of our next research.

We have tried our best to improve the manuscript and have made some changes in the manuscript. We marked with red where we made the changes. These changes do not affect the content and scope of the manuscript. We hope that the correction we have made will meet this requirement.

Thank you for your consideration. I look forward to hearing from you soon.

With best regards,

Prof. Jing He

College of Forestry,

Gansu Agricultural University,

Lanzhou 730030,

China.

This manuscript is a resubmission of an earlier submission. The following is a list of the peer review reports and author responses from that submission.

Round 1

Reviewer 1 Report

Major:

1) I recommend that the authors should use some help of a native English speaker or send the Ms to an English Editing Service that proofreads scientific writing.

2) Figure 1, 2, 3, 5, 6: increase font size.

3) Figure 1, 2: present as histograms

4) Authors should improve all legends for figures, e.g. used abbreviations, used statistical treatment, used fungal strains and plants.

5) Authors should decrease the figure number, e.g. by combination Fig. 5 and 6.

6) Can the authors provide photos of the root rot of wolfberry, maybe as additional material

Minor:

7) Line 9 in Abstract: authors should include the Latin name of the used wolfberry.

8) Line 75-76: “M. robertsii (HYC-7) and F. solani were isolated, screened and identified by forest Protection Laboratory, College of Forestry, Gansu Agricultural University.” Where are these strains stored?

9) Line 78: “distilled water (1000 mL).” correct to “distilled water (up to 1 L).”.

10) Line 82-85: include manufacturers of reagents, their purity.

11) Line 87-88: “1 mL spore suspension (1×107 mL-1) of 87 HYC-7 strain” - how was the number of spores measured?

12) Line 90-91: “was centrifuged at 4°C and 10000 rpm” - what kind of centrifuge or g (rcf)?

13) Line 92: “PDA medium” reference for this medium.

14) Line 93: “fungus cake” – explain.

15) Line 325-326: “and different letters indicate significant differences at P<0.05.” differences compared with?

I recommend that the authors should use some help of a native English speaker or send the Ms to an English Editing Service that proofreads scientific writing.

Author Response

Response to Reviewer 1 Comments

Major:

Point 1: I recommend that the authors should use some help of a native English speaker or send the Ms to an English Editing Service that proofreads scientific writing.

Response 1: Thanks for your suggestion. We have made moderate revisions to the manuscript in terms of language and grammar, as well as some sentences reconstruction and word choice changes for accuracy and improved clarity. The manuscript has undergone English language editing by MDPI (english-67148), and the grammar, spelling and punctuation have all been updated and checked.

Point 2: Figure 1, 2, 3, 5, 6: increase font size.

Response 2: We have increased the font size of Figure 1, 2, 3, 5, 6.

Point 3: Figure 1, 2: present as histograms.

Response 3: We have changed Figure 1 and Figure 2 to histograms as your suggestion.

Point 4: Authors should improve all legends for figures, e.g. used abbreviations, used statistical treatment, used fungal strains and plants.

Response 4: We have improved all figures legends in revised manuscript and marked in red.

Point 5: Authors should decrease the figure number, e.g. by combination Fig. 5 and 6.

Response 5: We have combined Figure 5 and Figure 6 together as your suggestion.

Point 6: Can the authors provide photos of the root rot of wolfberry, maybe as additional material.

Response 6: Unfortunately, we did not take photos of root rot of wolfberry during the experiment, but we counted the decay incidence and disease index of wolfberry root rot at different time and showed the results in the form of histogram (Fig. 6).

Minor:

Point 7: Line 9 in Abstract: authors should include the Latin name of the used wolfberry.

Response 7: We have added the Latin name ”Lycium barbarum” to the used wolfberry.

Point 8: Line 75-76: “M. robertsii (HYC-7) and F. solani were isolated, screened and identified by forest Protection Laboratory, College of Forestry, Gansu Agricultural University.” Where are these strains stored ?

Response 8:  M. robertsii (HYC-7) and F. solani were also stored at Forest Protection Laboratory, College of Forestry, Gansu Agricultural University, and we have added relevant information in the revised manuscript as below.

The isolation, screening, identification, and preservation of M. robertsii (HYC-7) and F. solani were carried out at the Forest Protection Laboratory, College of Forestry, Gansu Agricultural University.   

Point 9: Line 78: “distilled water (1000 mL).” correct to “distilled water (up to 1 L).”.

Response 9: “distilled water (1000 mL).” has been changed to “distilled water (up to 1 L).” in line 78.

Point 10: Line 82-85: include manufacturers of reagents, their purity.

Response 10: We have added the manufacturer of the reagent and its purity as below.

Mannitol, glucose, maltose, lactose, and soluble starch were selected as carbon sources to replace sucrose, and beef extract, yeast extract paste, L-glutamic acid, carbamide and peptone (purity≥99.7%, Tianjin Guangfu Fine Chemical Research Institute) were selected as nitrogen sources to replace sodium nitrate in the basic medium.

Point 11: Line 87-88: “1 mL spore suspension (1×107 mL-1) of HYC-7 strain” - how was the number of spores measured ?

Response 11: We used a hemocytometer to adjust the spore suspension concentration to 1×107 mL-1.

Point 12: Line 90-91: “was centrifuged at 4°C and 10000 rpm” - what kind of centrifuge or g (rcf) ?

Response 12: “rpm” have been changed to “g” as your suggestion.

The fermentation broth was centrifuged at 4 °C and 9900×g for 20 min to obtain the supernatant, and then filtered through a 0.22 μm microporous membrane for later use.

Point 13: Line 92: “PDA medium” reference for this medium.

Response 13:  We have added the reference for PDA midum in line 92.

Point 14: Line 93: “fungus cake” – explain.

Response 14: A plate of F. solani cultured for 7 days was taken, and a sterile puncher was used to punch the edge of the colony to obtain a medium round block with mycelium as a fungus cake for subsequent inoculation.

Point 15: Line 325-326: “and different letters indicate significant differences at P<0.05.” differences compared with ?

Response 15: In Figure 5, a significant difference analysis was conducted between the treatment and control at the same time point. We have revised the figure caption as below.

Line 325-326: Vertical bars represent standard error (±SE) and asterisks indicate significant differences between the treatment and the control at the same time (P<0.05).

We have tried our best to improve the manuscript and have made some changes in the manuscript. We marked with red where we made the changes. These changes do not affect the content and scope of the manuscript. We hope that the correction we have made will meet this requirement.

Thank you for your consideration. I look forward to hearing from you soon.

With best regards,

Prof. Jing He

College of Forestry,

Gansu Agricultural University,

Lanzhou 730030,

China.

Reviewer 2 Report

Authors proved to have a strange concept of biocontrol, when they decided to assess the capacity by their strain of M. robertsii to contrast root rot induced by Neocosmospora solani (this is the current name of the pathogen) on excised root fragments! Indeed, the concept of 'disease' can only be referred to a living organism, and evaluation of the biocontrol potential against a plant pathogen can only be done on living plants! Definitely, I think there is no need to find other arguments to lean for rejection of this manuscript.

The English style is quite approximate, and sometimes it is difficult to understand the sense of certain statements; particularly, in the M&M section.

Author Response

Response to Reviewer 2 Comments

Comments and Suggestions for Authors:

Point 1: Authors proved to have a strange concept of biocontrol, when they decided to assess the capacity by their strain of M. robertsii to contrast root rot induced by Neocosmospora solani (this is the current name of the pathogen) on excised root fragments! Indeed, the concept of 'disease' can only be referred to a living organism, and evaluation of the biocontrol potential against a plant pathogen can only be done on living plants! Definitely, I think there is no need to find other arguments to lean for rejection of this manuscript.

Response 1: Thanks for your suggestion. Root rot caused by Fusarium solani is an important soil-borne disease in the planting and cultivation of Wolfberry. The disease is difficult to detect before the plants show obvious symptoms. Therefore, in order to quickly and visually observed the decay of root rot, we used fresh root tissue for treatment and inoculation in this study.

Comments on the Quality of English Language

Point 2: The English style is quite approximate, and sometimes it is difficult to understand the sense of certain statements; particularly, in the M&M section. 

Response 2: Thanks for your suggestion. We have made moderate revisions to the manuscript in terms of language and grammar, as well as some sentences reconstruction and word choice changes for accuracy and improved clarity. The manuscript has undergone English language editing by MDPI, and the grammar, spelling and punctuation have all been updated and checked.

We have tried our best to improve the manuscript and have made some changes in the manuscript. We marked with red where we made the changes. These changes do not affect the content and scope of the manuscript. We hope that the correction we have made will meet this requirement.

Thank you for your consideration. I look forward to hearing from you soon.

With best regards,

Prof. Jing He

College of Forestry,

Gansu Agricultural University,

Lanzhou 730030,

China.

Reviewer 3 Report

Optimization of fermentation conditions of Metarhizium robertsii and its biological control of wolfberry root rot disease

The authors investigated the conditions for which M. robertsii would produce secondary metabolites that were antagonistic toward Fusarium solani. The study is important and needed as many fungal species can be used as biocontrol agents against other fungal and insect pathogens. The figures are well done. The main weakness of the manuscript is the grammar and organization, as well as some statements which need to be more speculative.

Grammar examples:

Abstract:

Lines 20-21: “In conclusion, M. robertsii has a certain biological control effect on root rot of wolfberry.” The sentence is awkward and vague.

Introduction:

Line 27: please consider changing to. “Shaanxi, and Qinghai provinces in China”

Line 36-38: “The current control of root rot is mostly chemical methods, but there are problems such as environmental pollution, high cost and the emergence of drug-resistant strains.”  The sentence is awkward and not sure about drug-resistance fungal strains. Chemical or fungicide resistance might be a better word choice.

Line 41-42: Please italicize all scientific names.

Line 43: “the plant growth” is award as written.

Line 45: Spell out first words in a sentence:  Consider: Metarhizium robertsii

Materials and methods: This section was what was done in the past so it should be written in past tense, not present tense.

Line 75: Spell out the first word of a sentence. Also, how were they screened and identified? Morphology? Molecular identification or both?

Lines 92-93, Lines 102-109; Lines 111-116 are award as written.

Line 90 and Line 121: when the authors write about centrifuge RPM, they should put the value in gravitational force because different centrifuges have different gravitational forces at the same rpm.

Line 179-180: “After the reaction solution is transparent, measured the absorbance value at 620 nm.” The sentence is awkward as written.

Line 190: Spell out Fusarium as the first word in a sentence.

Results and Discussion: Same issues throughout.

Overall organization issues:

The introduction is lacking information, particularly information about why the oxidative, protein, and sugar test were completed. A lot of this information is put in the discussion but would likely make the reader understand why they were done before the end of the manuscript. Consider moving some of the more background information in the discussion to the introduction.

Results should be in the past tense, please revise. In addition, the results have too much material and methods about the background of the statistics and most of this section can be condensed for clarification. Lines 320-322; Lines 341-343: these sentences are more of an interpretation and should be in the discussion.

Discussion: Lines 363-366: These sentences are in the template and not part of the manuscript. Lines 376-380: these sentences reiterate the results. Please limit repetition of repeating the results. Lines 427-429: The authors did not test this directly so it should be more of a speculative sentence.

The manuscript has some grammar and sentence structure issues which can be address by the authors upon revision.

Author Response

Response to Reviewer 3 Comments

Comments and Suggestions for Authors:

The authors investigated the conditions for which M. robertsii would produce secondary metabolites that were antagonistic toward Fusarium solani. The study is important and needed as many fungal species can be used as biocontrol agents against other fungal and insect pathogens. The figures are well done. The main weakness of the manuscript is the grammar and organization, as well as some statements which need to be more speculative. 

Response : Thank you for your positive comments and valuable suggestions. We have made moderate revisions to the manuscript in terms of language and grammar, as well as some sentences reconstruction and word choice changes for accuracy and improved clarity. The manuscript has undergone English language editing by MDPI (english-67148), and the grammar, spelling and punctuation have all been updated and checked. A point-by-point response is provided as below.

Grammar examples:

Abstract:

Point 1: Lines 20-21: “In conclusion, M. robertsii has a certain biological control effect on root rot of wolfberry.” The sentence is awkward and vague.

Response 1: We have revised the sentence as suggested below.

Line 22-23: These results indicate that M. robertsii could be used as a biological control agent in wolfberry root rot disease management.

Introduction:

Point 2: Line 27: please consider changing to. “Shaanxi, and Qinghai provinces in China”

Response 2: We have changed "Shaanxi, Qinghai provinces in China" to "Shaanxi, and Qinghai provinces in China" in the introduction.

Line 28-30: As a medicinal and functional food, wolfberry (Lycium barbarum) has a long history of planting and cultivation, and is widely planted in the Nei Monggol, Gansu, Ningxia, Shaanxi, and Qinghai provinces in China.

Point 3: Line 36-38: “The current control of root rot is mostly chemical methods, but there are problems such as environmental pollution, high cost and the emergence of drug-resistant strains.”  The sentence is awkward and not sure about drug-resistance fungal strains. Chemical or fungicide resistance might be a better word choice.

Response 3: We have deleted the sentence as suggested.

Point 4: Line 41-42: Please italicize all scientific names.

Response 4: We have italicized all the scientific names and checked the full text.

Point 5: Line 43: “the plant growth” is award as written.

Response 5: We have revised the sentence as suggested below.

Line 43-46: Trichoderma harzianum, Rhizobium japonicum, and T. atroviridae treatments significantly reduce the incidence and severity of peanut and soybean root rot caused by F. solani, and also show good plant growth promotion effects.

Point 6: Line 45: Spell out first words in a sentence:  Consider: Metarhizium robertsii.

 Response 6:  M. robertsii” has been changed to “Metarhizium robertsii” in line 52.

Materials and methods:

Point 7: This section was what was done in the past so it should be written in past tense, not present tense.

Response 7: We revised the manuscript in terms of tense, and made some sentences reconstruction and word selection changes to improve accuracy and clarity.

Point 8: Line 75: Spell out the first word of a sentence. Also, how were they screened and identified? Morphology? Molecular identification or both?

Response 8: We have spelled out the first word in line 75. In our previous study, the HYC-7 was identified as Metarhizium robertsii by morphological and molecular biological methods (Zhang et al., 2020).

Zhang, X. Y.; He, J.; Hou, C. X.; Zhang, S. H. Screening and identification of antagonistic strains of wolfberry root rot. Acta Agric Zhejiangensis202032, 858-865. DOI: 10. 3969 /j. issn. 1004-1524. 2020. 05. 14

Point 9: Lines 92-93, Lines 102-109; Lines 111-116 are award as written.

Response 9: We have rewritten these sentences as suggested below.

Lines 103-105: The fermentation broth was centrifuged at 4°C and 9900 × g for 20 min to obtain the supernatant, and then filtered through a 0.22 μm microporous membrane for later use. The fermentation broth was mixed with the PDA medium at a ratio of 30% [8]. After cooling, the fungus cake of F. solani was inoculated in the center of the plate and cultured at 28°C in the dark.

Lines 115-123: Under the basic conditions of temperature (28℃), pH (6.0), inoculation amount (1mL), loaded liquid (60 mL), and rotational speed (160 rpm), we kept the other conditions constant and only changed one condition to conduct a single-factor test. Set the following conditions, temperature: 20, 22, 25, 28, and 30℃; pH: 5.0, 6.0, 7.0, 8.0, 9.0, and 10.0; inoculation amount: 0.5, 1, 1.5, 2, 2.5, and 3 mL; loaded liquid: 40, 50, 60, 70, and 80 mL; rotational speed: 120, 140, 160, 180, and 200 rpm. Six replicates of each treatment were created and incubated for 5 d in a constant-temperature shaker; then, the inhibition rate was determined using the fermentation broth according to the method in 2.2.

Lines 125-130: On the basis of the single-factor test, the three parameters of pH (A), rotational speed (B), and temperature (C) were optimized. The response surface test scheme was designed using the BOX-Design-Expert 8.0.6 software (Table S1). The center combination design was carried out to determine the optimal fermentation conditions. The results are shown in Table S2, taking into account the experimental error; 5 sets of central tests were set up, and a total of 17 tests were tested.

Point 10: Line 90 and Line 121: when the authors write about centrifuge RPM, they should put the value in gravitational force because different centrifuges have different gravitational forces at the same rpm.

Response 10: “rpm” have been changed to “g” as your suggestion.

Point 11: Line 179-180: “After the reaction solution is transparent, measured the absorbance value at 620 nm.” The sentence is awkward as written.

Response 11: We have revised the sentence as suggested below.

Line 179-180: After the reaction solution become transparent, the absorbance value was determined at 620 nm.

Point 12: Line 190: Spell out Fusarium as the first word in a sentence.

Response 12: We have spelled out the full name of the first word.

Line 224-226: Fusarium solani with a concentration of 1×106 spores/mL were sprayed uniformly on the surface of the root tissues after 24 h and then cultured at 28℃ in the dark.

Point 13: Results and Discussion: Same issues throughout.

Response 13: In the results and discussion section, we have modified some of the issues you mentioned in revised manuscript and marked in red.

Overall organization issues:

Point 14: The introduction is lacking information, particularly information about why the oxidative, protein, and sugar test were completed. A lot of this information is put in the discussion but would likely make the reader understand why they were done before the end of the manuscript. Consider moving some of the more background information in the discussion to the introduction.

Response 14: We have added some background information from the discussion to the introduction in revised manuscript as your suggested and marked in red.

Line 53-60: Antioxidant systems play an important role in ROS scavenging, and superoxide dismutase (SOD), catalase (CAT), and peroxidase (POD) are important antioxidant enzymes in pathogenic fungi [12]. The inhibition of antioxidant enzyme activity may disrupt the balance of ROS metabolism, thus affecting the growth and pathogenicity of pathogenic fungi. MDA is one of the indicators to measure the degree of oxidative stress, which can reflect the degree of fungal membrane lipid peroxidation. Soluble protein and soluble sugar could act as measures for the level of protein damage and cell carbon metabolism.

Point 15: Results should be in the past tense, please revise. In addition, the results have too much material and methods about the background of the statistics and most of this section can be condensed for clarification. Lines 320-322; Lines 341-343: these sentences are more of an interpretation and should be in the discussion.

Response 15: We have revised the tense in the results and integrated the material and methods about the background of the statistics from results 3.3 into material and methods section. In addition, sentences in the results that resemble interpretation are placed in the discussion as your suggested and marked in red in revised manuscript.

Line 124-148: 

2.4 Response surface optimization test

On the basis of the single-factor test, the three parameters of pH (A), rotational speed (B), and temperature (°C) were optimized. The response surface test scheme was designed using the BOX-Design-Expert 8.0.6 software (Table S1). The center combination design was carried out to determine the optimal fermentation conditions. The results are shown in Table S2, taking into account the experimental error; 5 sets of central tests were set up, and a total of 17 tests were tested. Design-expert 10 software was used to perform multiple regression fitting analysis on the data in the table, and the inhibition rate (Y) of the fungus fermentation broth was obtained using the pH (A), rotational speed (B, rpm) and temperature (C, °C). The quadratic multinomial regression equation is:

Y=51.71-3.09 A-0.10 B+3.37 C+2.96 AB+0.45AC+0.80 BC-4.38 A2-8.39 B2-7.51 C2.

2.5 Regression equation fitting and analysis of variance

Variance analysis and the significant difference test were conducted for the regression model, and the results are shown in Table S3. The regression of the model was significant (P<0.0001). The loss of quasi-item P=0.1179 > 0.05 was not significant, indicating that the test results were consistent with the model, while other unknown factors had little interference in the test results; the model was suitable and the fitting degree was high so the establishment of the regression model had a certain guiding significance. At the same time, the first term A, C, AB and the second term A2, B2, C2 all had significant antifungal activity. The regression coefficient R2=0.9777; this showed that the correlation of the model was high. The F value represents the importance of each influencing factor to the test index. The larger the F value is, the stronger the influence of the factor on the inhibition rate. The results showed that the influence of 3 factors on the inhibition rate was in the order of temperature (C, °C) > pH (A) > rotational speed (B, rpm).

Point 16: Discussion: Lines 363-366: These sentences are in the template and not part of the manuscript. Lines 376-380: these sentences reiterate the results. Please limit repetition of repeating the results. Lines 427-429: The authors did not test this directly so it should be more of a speculative sentence.

Response 16: We have deleted the first sentences and simplified some of the repeated results in the discussion as suggested below, and marked in red in revised manuscript.

Lines 372-374: The response surface results indicated that temperature, pH, and rotational speed had a greater effect on the antifungal activity of M. robertsii fermentation broth.

Lines 415-42 : In this study, we found that the HYC-7 fermentation broth treatment promoted the accumulation of MDA and accelerated the leakage of soluble protein and the decrease in soluble sugar content. Therefore, we inferred that the HYC-7 fermentation broth caused serious damage to cell membrane and the leakage of soluble protein, and inhibited the carbon metabolism of F. solani, limiting its normal life metabolism, thus effectively inhibiting the growth of F. solani.

We have tried our best to improve the manuscript and have made some changes in the manuscript. We marked with red where we made the changes. These changes do not affect the content and scope of the manuscript. We hope that the correction we have made will meet this requirement.

Thank you for your consideration. I look forward to hearing from you soon.

With best regards,

Prof. Jing He

College of Forestry,

Gansu Agricultural University,

Lanzhou 730030,

China.

Round 2

Reviewer 1 Report

Minor:

Previously remark:

“4) Authors should improve all legends for figures, e.g. used abbreviations, used statistical treatment, used fungal strains and plants.”

- Authors should check the legends for figures again, e.g. Fig. 5 – explain CK, HYC-7, SOD, CAT, POD, MDA; Fig. 6 – explain CK, HYC-7, etc.

Minor editing of English language required

Author Response

Minor:

Point 1: “4) Authors should improve all legends for figures, e.g. used abbreviations, used statistical treatment, used fungal strains and plants.”

- Authors should check the legends for figures again, e.g. Fig. 5 – explain CK, HYC-7, SOD, CAT, POD, MDA; Fig. 6 – explain CK, HYC-7, etc.

Response 1: We have defined HYC-7 in line 84 and added more detailed information in Fig. 5 and Fig. 6 legends as your suggestion. In addition, we have also changed “CK” to “Control” in Fig. 5 and Fig. 6.

Comments on the Quality of English Language

Point 2: Minor editing of English language required.

Response 2: We have made moderate revisions to the manuscript and the grammar, spelling and punctuation have all been updated and checked.  

We have tried our best to improve the manuscript and have made some changes in the manuscript. We marked with red where we made the changes. These changes do not affect the content and scope of the manuscript. We hope that the correction we have made will meet this requirement.

Thank you for your consideration. I look forward to hearing from you soon.

With best regards,

Prof. Jing He

College of Forestry,

Gansu Agricultural University,

Lanzhou 730030,

China.

Reviewer 3 Report

The Authors have made good progress on the manuscript. It is much easier to read, the organization has improved, and the methods are clearer. Comment below.

Key Words: All words in the title are essentially key words so its best not to repeat them in the key words section as it will limit the article in literature search. I would suggest removing the words that match title and replace with others.

Line 49: Please capitalize Lobesia and italicize Lobesia botrana.

Line 55: Please spell out MDA.  Malondialdehyde (MDA)

Line 123, Line131, line 141: These are results and are awkward in the methodology section. Suggest removing if this is repeated in the results section. If they are not, suggest moving to the results section. Also, make sure to cite the tables in the results section if these are removed.

Line 187: Suggest defining MDA on line 55 and spelling out the word in line 187 as it’s the first word of a sentence.

Section 3.3: Explaining how to interpret the response surface diagram in the results appears confusing as its not a result. This would be better in the discussion, or perhaps. Part of Figure 3 caption so the reader can see the images and understand how to interpret.

Line 279 and 280: Suggest removing: This showed that. Just state the model had a food simulation effect.

Author Response

Comments and Suggestions for Authors:

The Authors have made good progress on the manuscript. It is much easier to read, the organization has improved, and the methods are clearer. Comment below.

Response : Thank you for your positive comments and valuable suggestions. We have made moderate revisions to the manuscript and a point-by-point response is provided as below.

Point 1: Key Words: All words in the title are essentially key words so its best not to repeat them in the key words section as it will limit the article in literature search. I would suggest removing the words that match title and replace with others.

Response 1: The keywords Metarhizium robertsii; Fermentation conditions; Wolfberry; Root rot; Fusarium solani” have been changed to “Endophytic fungus; Lycium barbarum; Disease control; Fusarium solani; Antifungal mechanism” as your suggestion.

Point 2: Line 49: Please capitalize Lobesia and italicize Lobesia botrana.

Response 2: “lobesia botrana” has been changed to “Lobesia botrana” in line 49.

Point 3: Line 55: Please spell out MDA.  Malondialdehyde (MDA).

Response 3: We have spelled out “Malondialdehyde (MDA)” in line 55 as your suggestion.

Point 4: Line 123, Line131, line 141: These are results and are awkward in the methodology section. Suggest removing if this is repeated in the results section. If they are not, suggest moving to the results section. Also, make sure to cite the tables in the results section if these are removed.

Response 4: We have moved some pf the results you mentioned to the results section.

Point 5: Line 187: Suggest defining MDA on line 55 and spelling out the word in line 187 as it’s the first word of a sentence.

Response 5: We have defined MDA on line 55 and spelled out the word in line 187 as your suggestion.

Point 6: Section 3.3: Explaining how to interpret the response surface diagram in the results appears confusing as its not a result. This would be better in the discussion, or perhaps. Part of Figure 3 caption so the reader can see the images and understand how to interpret.

Response 6: We have moved the sentences to the caption of Figure 3 as your suggestion.

Point 7: Line 279 and 280: Suggest removing: This showed that. Just state the model had a food simulation effect.

Response 7: We have removed the sentences as your suggestion.

We have tried our best to improve the manuscript and have made some changes in the manuscript. We marked with red where we made the changes. These changes do not affect the content and scope of the manuscript. We hope that the correction we have made will meet this requirement.

Thank you for your consideration. I look forward to hearing from you soon.

With best regards,

Prof. Jing He

College of Forestry,

Gansu Agricultural University,

Lanzhou 730030,

China.
